# Photosynthetic and Physiological Responses to Combined Drought and Low–Temperature Stress in *Poa annua* Seedlings from Different Provenances

**Juanxia Li [1], Xiaoming Bai [1,2,*], Fu Ran [1], Ping Li [1], Mahran Sadiq [1] and Hui Chen [1]**

[1] College of Grassland Science, Gansu Agricultural University, Lanzhou 730070, China; ljx826824@163.com (J.L.)
[2] Key Laboratory of Grassland Ecosystem (Ministry of Education), Lanzhou 730070, China
* Correspondence: baixm@gsau.edu.cn

**Abstract:** Combined drought and low–temperature stress is a crucial factor affecting turfgrass establishment and limiting the sustainability of the turfgrass industry in drought– and cold–prone regions. In this context, we evaluated the effects of regular watering (the soil water content was 80% of the maximum water–holding capacity of the field) at room temperature (25 °C) and combined drought (the soil water content was 30% of the maximum water–holding capacity of the field) and low–temperature (0 °C) stress on the morphology, photosynthesis, and physiology of wild *Poa annua* seedlings from different provenances ('PA', 'WY', 'NX' and 'YC'). Results indicated that the combined drought and low–temperature stress changed the morphological and growth indicators of seedlings in four provenances to different extents. Moreover, combined drought and low–temperature stress reduced the net photosynthetic rate (Pn), stomatal conductance (Gs), transpiration rate (Tr), water use efficiency (WUE), and chlorophyll content in seedlings from four provenances. However, intertemporal $CO_2$ concentration (Ci), relative electrical conductivity (REC), the contents of malondialdehyde (MDA), proline (Pro), soluble sugars (SS), the superoxide anion ($O_2^{\bullet-}$) production rate, the contents of hydrogen peroxide ($H_2O_2$) and hydroxyl radical (·OH), the activities of superoxide dismutase (SOD), peroxidase (POD), catalase (CAT), and ascorbate peroxidase (APX) were all increased. The increase in 'PA' was much greater than that in 'NX.' The comprehensive evaluation results showed that the order of combined drought and low–temperature resistance of seedlings from the four provenances was 'PA' > 'YC' > 'WY' > 'NX', which corresponded to the order of the morphological damage symptoms. In conclusion, 'PA' may maintain stronger combined drought and low–temperature resistance by improving the cellular water absorption and retention capacity, enhancing the function of the antioxidant defense system, and maintaining the integrity of the cell membrane, which is a crucial germplasm resource for breeding combined drought and low–temperature resistance in *Poa annua*.

**Keywords:** *Poa annua*; provenance; combined drought and low–temperature stress; growth characteristics; physiology and biochemistry; principal component





## 1. Introduction

In nature, plants are exposed to the combined effects of multiple abiotic stresses throughout their life cycle, posing severe threats to plant growth and productivity [1–4]. Currently, most studies have assessed plant response to each single stress at home and abroad [5,6]. However, under field conditions, the superposition of multiple abiotic stresses is inevitable, which may be more severe than the effects of each single stress. Thus, a new approach based on studying overlapping impacts of abiotic stresses seems very important for analyzing plant responses under environmental conditions [7,8]. Drought and low–temperature are the most common abiotic factors limiting plant growth and distribution, and both often co–occur [9]. Previous studies have shown that drought and low–temperature hinder plant growth and development by increasing antioxidant activity, reactive oxygen species, and lipid peroxidation, impairing membrane stability, decreasing chlorophyll (Chl) biosynthesis,

and disrupting the photosynthetic system [10–12]. However, these response mechanisms vary depending on plant species, provenances, and severity of stress. Generally speaking, species or provenances from bad environments perform better in terms of survival and growth than species or provenances with better backgrounds [13–15]. The successful establishment of plants in new locations depends not only on the ability to survive adversity but also on their physiological capability to face mutant environments [16]. Therefore, it is also crucial to select and utilize species and provenances that are potentially resistant to environmental conditions and mutations.

Combined stress usually limits plant respiration and photosynthesis [17], breaking the balance of osmoregulatory substances, antioxidant enzymes, phytohormones, and other substances in plant cells, thereby affecting plant growth, development, and physiological metabolism [18,19]. Under combined drought and low–temperature stress, plants produce excess reactive oxygen species (ROS) and cause oxidative damage [20]. Furthermore, ROS can damage a variety of cellular components, such as proteins and lipids, and unrestricted damage will eventually lead to cell death [21]. However, in long–term species evolution, plants can develop a range of enzymatic and non–enzymatic antioxidant systems in response to oxidative stress to scavenge excess ROS and avoid their toxicity to cellular biomolecules [22]. Moreover, plants can avoid damage to cellular structures by degrading or retaining chlorophyll under stress, thus avoiding the overproduction of ROS in the electron transport chain of plant chloroplasts [23–25]. However, plant responses to combined drought and low–temperature stress vary by species and provenances, laying a foundation for plant improvement and breeding.

*Poa annua* L. is an annual cool–season turfgrass in the Gramineae family with excellent traits, such as short life history, strong reproduction, trampling resistance, pruning resistance, strong resistance, and ease of establishment. It is often used as an excellent short–term ornamental lawn in the winter and spring or as a cross–seeding material to prolong the greening period of the lawn. Recently, it has been widely noticed as a new type of turfgrass species [26], but it can also be affected by extreme weather. In the northern area of China, especially in the northwest region, due to scarce rainfall, water scarcity, and cold and dry winters, resulting in turfgrass safety overwintering and return to green on the schedule are hindered, which not only causes enormous economic losses to the turfgrass industry but also seriously affects the quality of lawn establishment. Combined drought and low–temperature stress may have synergistic or antagonistic effects on plants and can not be regarded as a simple superimposed effect. As far as we know, the effects of combined drought and low–temperature stress on plant growth are still scarce. Moreover, the magnitude of plant resistance may also vary depending on provenances. Thus, we investigated the changes in morphological and physiological biochemical parameters of wild *Poa annua* from four provenances under sufficient water (the soil water content was 80% of the maximum water–holding capacity of the field) at room temperature (25 °C) and combined drought (the soil water content was 30% of the maximum water–holding capacity of the field) and low–temperature (0 °C), and compared the combined drought and low–temperature resistance among different provenances to better screen for materials with combined drought and low–temperature resistance, providing a theoretical basis for selecting of *Poa annua* germplasm suitable for cultivation in arid and low–temperature regions and cultivation of new varieties.

## 2. Materials and Methods

### 2.1. Plant and Culture

Seeds of *Poa annua* from four provenances were obtained from May to July 2021 in Gansu and Qinghai Provinces, China, and named after the provenances. Furthermore, the geographical locations and climatic conditions of provenances are shown in Table 1 and Figure 1. The experiment was conducted in Gansu Agricultural University, Gansu Province, China (36°5′ N, 103°34′ E), with an average annual temperature of 10.3 °C and average annual precipitation of 350 mm. The seeds were sown in plastic pots in June 2022.

The cultivated substrate was farmland soil, sand, sheep manure, and organic nutrient soil (7:1:1:1; $v/v$) [27,28]. Each pot was filled with 1.8 kg of the mixed substrate, with a sowing rate of 8 g·m$^{-2}$, and cultivated under natural conditions. Furthermore, the position of the pots was moved periodically to reduce differences in the microenvironment.

**Table 1.** Locations and climatic data for sites of seed sampled in Gansu and Qinghai provinces of China.

| No | Provenances | Latitude and Longitude | Altitude | Mean (Min–Max) Temperature | Mean (Min–Max) Precipitation | Habitats |
|---|---|---|---|---|---|---|
| PA | Ping'an District, Xining City, Qinghai Province | 102°10′58″ E 36°25′37″ N | 2210 | −9.5~19 °C | 1.64~82.36 | Green belt |
| WY | Weiyuan County, Dingxi City, Gansu Province | 107°32′18″ E 35°27′29″ N | 2077 | −5.5~19 °C | 2.41~99.36 | In front of and behind the house |
| NX | Ning County, Qingyang City, Gansu Province | 107°55′41″ E 35°300′8″ N | 1337 | −3.5~24.5 °C | 4.13~112.72 | In front of and behind the house |
| YC | Yongchang County, Jinchang City, Gansu Province | 104°54′21″ E 34°10′41″ N | 1965 | −8.0~20.5 °C | 0.84~45.37 | Green belt |

Mean (min–max) temperature and (min–max) precipitation data were from Weather rp5 (https://rp5.ru/ (accessed on 8 June 2023)).

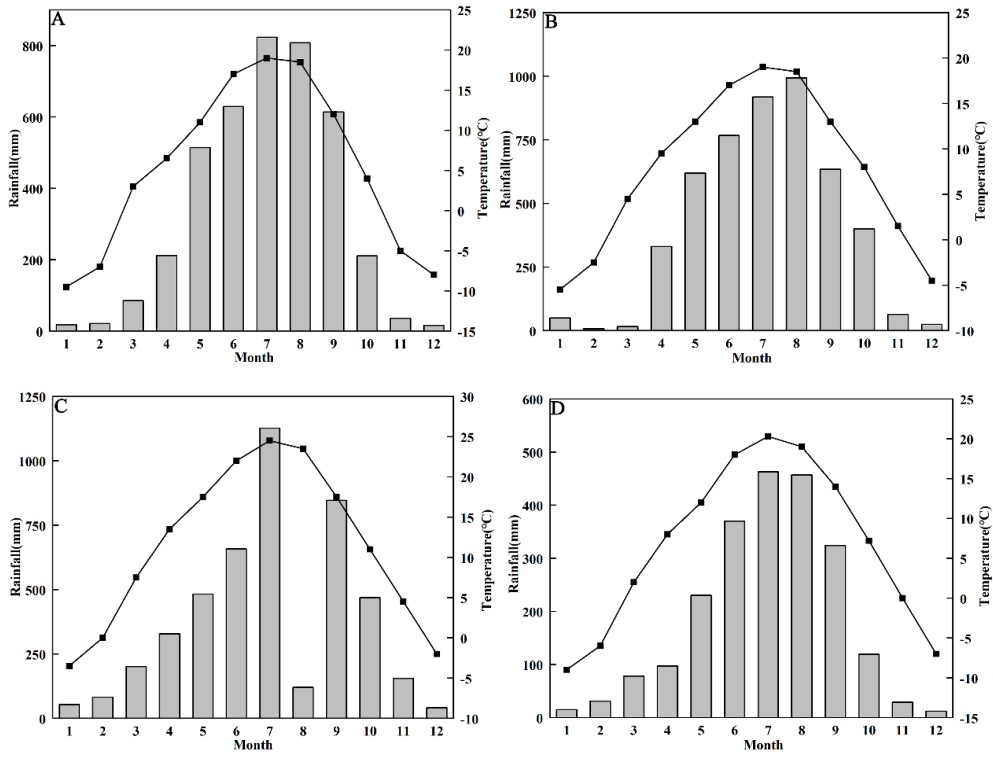

**Figure 1.** Climographs based on mean precipitation (bars) and mean temperature (black square) in 'PA' (**A**), 'WY' (**B**), 'NX' (**C**), and 'YC' (**D**), temperature and precipitation data are sourced from Weather rp5 (https://rp5.ru/ (accessed on 8 June 2023)).

### 2.2. Stress Treatment

Two–month–old seedlings were exposed to combined drought and low–temperature stress. The experiment was conducted in a completely randomized design. Seven days before the stress treatment, in order to maintain the same soil water content and temperature, we used the weighing method at room temperature (25 °C) every day; the soil water content of seedlings in all provenances should be 80% of the maximum water-holding capacity of the field. After that, *Poa annua* seedlings from all provenances were divided into control and stress groups, with three replicates of each treatment. Seedlings in the control group continued to be maintained at room temperature (25 °C), and the soil water content was at 80% of the maximum water–holding capacity of the field. The seedlings in the stress group stopped watering and gradually became dry naturally until the soil water content decreased to 30% of the maximum water–holding capacity of the field. Then, the seedlings were pre–cooled in a low–temperature incubator at 15 °C for 24 h, and the temperature was lowered to 0 °C at a rate of 5 °C/2 h, and then subjected to the 24 h of combined drought and low–temperature stress, and the soil water content of the seedlings was maintained at 30% of the maximum water–holding capacity of the field throughout the stress. The maximum water–holding capacity of the field was determined in the experiment using the stratified soil extraction method, and the soil water content was determined by the weighing method [29]. After the end of the stress, a portion of seedlings in each group was randomly selected for the determination of growth indexes, and another portion of middle and upper functional leaves was sampled for the determination of relevant physiological and biochemical indexes.

### 2.3. Growth Indicators

Before stress, nine uniformly growing seedlings (three per pot) were randomly picked from each group, and seedling height (SH), leaf length (LL), and leaf width (LW) were recorded, and leaf area (LA) was determined using a leaf area meter (LI–3100, LI–COR, Lincoln, NE, USA). Meanwhile, the fresh weight (FW) and saturated fresh weight (SFW) of the leaves were weighed using an electronic balance and then placed in an oven (DHG–9030, Shanghai Yiheng Scientific Instrument Co., Ltd., Shanghai, China) at 75 °C to dry until constant weight. They were weighed for dry weight (DW), and then the following calculations were performed using the formula: Leaf dry matter content (LDMC) = DW/FW. Relative water content of leaves (RWC) = (FW − DW)/(SFW − DW) × 100%. After stress treatment, the SH, LL, LW, LA, LDMC, and RWC of the seedlings were measured again. Moreover, photographs were taken to record the morphological changes of seedlings before and after stress.

### 2.4. Gas Exchange Parameters

Functional leaves with similar and healthy growth conditions at the mid–upper edges were selected for measurement in the morning (9:00 to 11:00 a.m. on sunny days). The net photosynthetic rate (Pn), intercellular $CO_2$ concentration (Ci), stomatal conductance (Gs), and transpiration rate (Tr) were determined with a portable photosynthesizer (Ciras–2, MA01913, Amesbury, MA, USA). Water use efficiency (WUE) was calculated as the ratio of Pn/Tr.

### 2.5. Chlorophyll Content

Leaf pigments were extracted using 95% (*v/v*) ethanol [30]. Chlorophyll a (Chl a) and chlorophyll b (Chl b) contents of the extracts were determined at an absorbance of 665 nm and 649 nm by UV–Vis spectrophotometer (Cary 60 UV–Vis, Agilent Technologies Inc., Santa Clara, CA, USA), and Chla/b and Chla + b contents were calculated.

### 2.6. Relative Electrical Conductivity and Malondialdehyde Content

Relative electrical conductivity (REC) was measured using the conductivity method [31]. The leaves were immersed in ultrapure water at a concentration of 0.1 g and subsequently

incubated at a temperature of 40 °C for a duration of 30 min. The conductivity ($R_1$) was measured using a conductivity meter (HI2314, Hanna Instruments, Woonsocket, RI, USA). Then, the sample was heated in boiling water for 15 min. The sample was allowed to cool before proceeding to measure its conductivity ($R_2$). REC was calculated as the ratio of $R_1/R_2$.

The content of malondialdehyde (MDA) was determined by the thiobarbituric acid (TBA) method [32]. Briefly, the leaves of the seedlings were extracted with 5% (*w/v*) trichloroacetic acid (McLean Biotechnology Co., Ltd., Shanghai, China). A total of 2 mL of 0.6% TBA (Beijing Solaybao Biotechnology Co., Ltd., Beijing, China) was added to each 2 mL supernatant. The mixture was heated in a water bath (HWS–26, Shanghai Yiheng Scientific Instrument Co., Ltd., Shanghai, China) at 100 °C for 30 min, after cooling naturally, then centrifuged (12,000 rpm/min) at 4 °C for 10 min, and the absorbance at 532 nm and 600 nm was recorded on a UV–Vis spectrophotometer.

### 2.7. Osmoregulatory Substances

Proline (Pro) content was determined using the ninhydrin method [33]. Briefly, seedling leaves were extracted with 3% (*w/v*) sulfosalicylic acid (Beijing Solaybao Bio–technology Co., Ltd., Beijing, China). A total of 1 mL of acidic ninhydrin (McLean Biotechnology Co., Ltd., Shanghai, China) and glacial acetic acid (Sinopharm, Beijing, China) was added to each 1 mL of supernatant. The mixture was heated at 100 °C for 30 min, cooling after, and then 2 mL of toluene was added in the middle of the extracted mixture. Finally, the absorbance was measured at 520 nm by a UV–Vis spectrophotometer.

The soluble sugar (SS) content was determined using the anthrone colorimetric method [34]. Seedling leaves were extracted with distilled water and then mixed with an–throne ethyl acetate (Sinopharm, Beijing, China) and concentrated sulfuric acid (Sinopharm, Beijing, China). The mixture was heated in boiling water for 1 min, and after cooling, the absorbance was measured at 630 nm by a UV–Vis spectrophotometer.

### 2.8. Reactive Oxygen Levels

The superoxide anion ($O_2^{\bullet-}$) production rate was determined by the hydroxylamine oxidation method [34]. The $H_2O_2$ content was determined with the acetone method [35]. Hydroxyl radical ($\cdot OH$) was determined using the salicylic acid method [36].

### 2.9. Antioxidant Enzyme Activity

The antioxidant enzyme activity was determined according to the method of Khalid et al. [37] and Nakano and Asada [38], and some improvements have been made. Briefly, 0.2 g of leaves were homogenized in 5 mL of pre–cooled phosphate buffer (50 mM, pH 7.8) and then centrifuged (5000 rpm/min) at 4 °C for 10 min; the supernatant was crude enzyme extract, and then the supernatant was collected to determine the activities of superoxide dismutase (SOD), peroxidase (POD), catalase (CAT) and ascorbic acid peroxidase (APX).

### 2.10. Statistical Analysis

Using the SPSS 20.0, the data were subjected to ANOVA and Duncan multiple comparisons. Two–way analyses of variance (ANOVA) were conducted to detect the effects of provenance and combined drought and low–temperature stress and their interactions. Origin 2021.0 was used for plotting figures. Canoco 5.0 was used to test the principal component analysis (PCA), screening the measured indices. The traits were comprehensively analyzed using the fuzzy mathematical membership function method to evaluate the combined drought and low–temperature tolerance of the *Poa annua* from different provenances. The value of the affiliation function was calculated as follows [39,40]:

$$R(X_i) = \frac{(X_i - X_{\min})}{(X_{\max} - X_{\min})} \tag{1}$$

$$R(X_i) = \frac{(X_{max} - X_i)}{(X_{max} - X_{min})} \tag{2}$$

$$U_j = \frac{1}{n} \sum_{i=1}^{n} R(X_i) \tag{3}$$

where $X_i$ is the measured value of the trait, and $X_{min}$ and $X_{max}$ are the minimum and maximum values of a certain trait of all tested materials, respectively. When the index is positively correlated with the combined drought and low–temperature resistance of *Poa annua*, use Formula (1) to calculate the membership function value. When the index is negatively associated with the combined drought and low–temperature resistance of *Poa annua*, use Formula (2) to calculate the membership function value. $U_j$ is the average value of the membership function of the traits measured for combined drought and low–temperature resistance of the *Poa annua*. In addition, the coefficient of genetic variation (CV%) for single traits of *Poa annua* under each treatment was calculated: (CV%) = SD/$\overline{X}$, where SD represents the standard deviation, and $\overline{X}$ represents the mean value of single traits among the materials under treatment.

## 3. Results

### 3.1. Changes in Morphological Characteristics of Seedlings from Four Provenances under Combined Drought and Low–Temperature Stress

The damage symptoms of four seedlings appeared gradually under combined drought and low–temperature stress. However, significant differences were noted across four provenances (Figure 2, Table 2). Four seedlings grew normally under regular watering at room temperature. However, after 24 h of combined drought and low–temperature stress, the upper leaves of four seedlings appeared to have symptoms of freezing spots, wilting, and curling; among them, the seedlings of the 'NX' showed the most severe damage symptoms and noticeable browning of the stem. The damage symptoms of the 'WY' and 'YC' seedlings were second only to 'NX', while the yellowing symptoms of the 'WY' seedlings were more severe. Compared with the other three seedlings, the stress damage symptoms of the 'PA' seedlings were the least severe. Therefore, the combined drought and low–temperature resistance of seedlings from four provenances can be preliminarily evaluated based on phenotypic symptoms as follows: 'PA' > 'YC' > 'WY' > 'NX.'

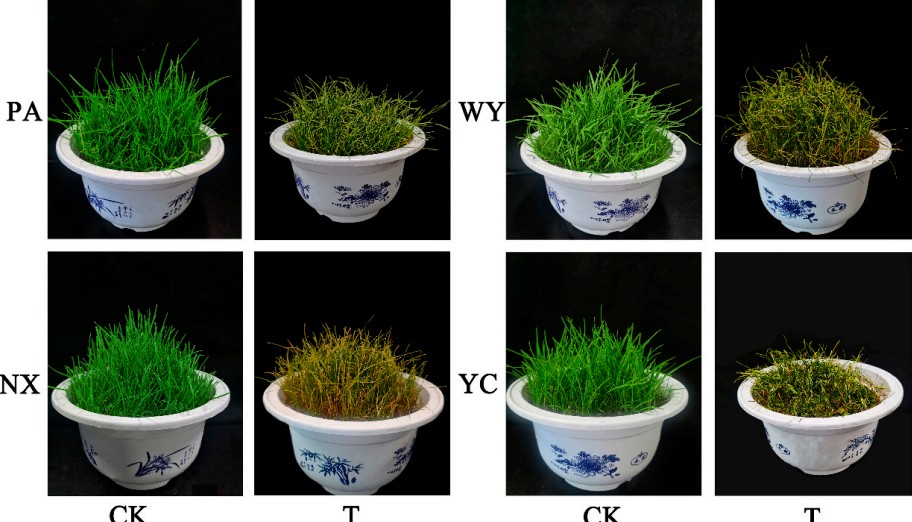

**Figure 2.** Comparison of morphological traits of seedlings from different provenances under normal watering at room temperature (CK) and combined drought and low–temperature stress (T). Scale bar: 5 cm.

**Table 2.** Morphological characteristics of *Poa annua* seedlings from four different provenances under combined drought and low–temperature stress.

| Treatments | Plant Morphological Characteristics | | | |
|---|---|---|---|---|
| | PA | WY | NX | YC |
| Normal watering at room temperature | The normal color of leaves and growth of the plant | The normal color of leaves and growth of the plant | The normal color of leaves and growth of the plant | The normal color of leaves and growth of the plant |
| Combined drought and low–temperature stress | Partial leaf margin wither and curl and frost spots at the upper of the plant | More leaves became yellow, withered, and curled, and partial stems turned brown | Most leaves became visibly yellow, wilted, and curled, and the stems turned dark brown | more leaves became yellow, withered, and curled, and appeared visibly frost spots |

### 3.2. Analysis of Variance (ANOVA) for Traits of Poa annua Seedlings

Analysis of variance (ANOVA) in response to provenance and combined drought and low–temperature stress proved highly significant differences ($p \leq 0.01$) in morphological, physiological, and biochemical traits (Table 3), which will help to screen combined drought and low–temperature resistant *Poa annua* germplasms. Furthermore, the interaction of provenances and combined drought and low–temperature stress also had significant differences, indicating that the resistance of *Poa annua* to the combined drought and low–temperature stress was controlled by its own genetic constitution, water and temperature, and interaction. Our results also suggested that the combined drought and low–temperature stress had a greater effect on 26 traits. The genetic differences were confirmed by coefficients of genetic variation of *Poa annua* from different provenances at the seedling stage (Table 3). The coefficients of genetic variation of seedlings from four provenances under adequate water at room temperature ranged from 7.57% [Chlorophyll a/b (Chla/b)] to 52.08% [(proline content)], and the coefficient of genetic variation (CV%) under combined drought and low–temperature stress ranged from 8.75% [ascorbate peroxidase (APX) activity] to 78.40% [stomatal conductance (Gs)].

**Table 3.** Factorial ANOVA of the provenance (P) and combined drought and low–temperature stress (D and C), their interactions (P × D and C), and genetic variation coefficient on seedlings from four provenances.

| Trait | Main Factors | | Interaction | Residual | CV (CK) (%) | CV (T) (%) |
|---|---|---|---|---|---|---|
| | P | D and C | P × D and C | | | |
| SH | 50.970 | 30.973 | 0.362 | 0.169 | 12.10 | 15.60 |
| LL | 124.396 | 20.485 | 0.387 | 0.323 | 22.43 | 22.59 |
| LW | 12.687 | 9.278 | 0.041 | 0.000 | 15.52 | 18.31 |
| LA | 72.376 | 16.680 | 0.715 | 0.048 | 36.18 | 36.77 |
| LDMC | 8.020 | 40.900 | 6.203 | 0.000 | 11.74 | 15.19 |
| RWC | 32.011 | 212.103 | 0.840 | 0.002 | 14.91 | 19.17 |
| Chla | 12.956 | 89.962 | 0.796 | 0.019 | 15.56 | 28.34 |
| Chlb | 8.630 | 146.149 | 8.744 | 0.004 | 7.59 | 17.45 |
| Chla/b | 11.545 | 14.241 | 4.794 | 0.011 | 7.57 | 22.66 |
| Chla + b | 12.553 | 119.859 | 1.980 | 0.036 | 12.32 | 20.53 |
| Pn | 46.355 | 907.791 | 36.506 | 0.194 | 30.05 | 31.58 |
| Ci | 101.019 | 133.648 | 9.993 | 278.722 | 28.89 | 12.90 |
| Gs | 94.973 | 169.174 | 2.341 | 205.007 | 29.93 | 78.40 |
| Tr | 6.941 | 231.797 | 7.205 | 0.101 | 10.10 | 35.03 |
| WUE | 42.502 | 244.928 | 16.487 | 0.025 | 32.68 | 29.76 |

**Table 3.** *Cont.*

| Trait | Main Factors | | Interaction | Residual | CV (CK) (%) | CV (T) (%) |
|---|---|---|---|---|---|---|
| | P | D and C | P × D and C | | | |
| REC | 2.021 | 277.194 | 1.526 | 0.006 | 23.05 | 13.48 |
| MDA | 2.672 | 301.750 | 6.197 | 3.285 | 10.55 | 9.59 |
| $O_2^{\bullet -}$ | 28.759 | 11.699 | 0.558 | 0.270 | 23.59 | 24.54 |
| $H_2O_2$ | 39.604 | 2372.459 | 41.105 | 0.231 | 14.29 | 10.84 |
| ·OH | 29.437 | 252.002 | 13.215 | 0.001 | 23.42 | 15.68 |
| Pro | 87.013 | 491.781 | 12.156 | 6.787 | 52.08 | 31.97 |
| SS | 2.820 | 48.431 | 3.772 | 0.026 | 23.14 | 13.48 |
| SOD | 127.748 | 1572.355 | 64.268 | 37.834 | 16.95 | 29.21 |
| POD | 91.875 | 2315.116 | 4.364 | 856.418 | 38.19 | 12.39 |
| CAT | 10.733 | 427.965 | 2.071 | 27.059 | 39.18 | 13.45 |
| APX | 18.537 | 729.297 | 1.943 | 6.741 | 33.43 | 8.75 |

The abbreviations of the corresponding indicators are shown in materials and methods. The same as below.

### 3.3. Effects of Combined Drought and Low–Temperature Stress on the Growth Characteristics of Different Seedlings

As shown in Figure 3, SH, LL, LW, LA, and RWC of seedlings from four provenances decreased, while LDMC increased. Compared with non–stressed seedlings, the RWC of the 'PA', 'WY', 'NX', and 'YC' seedlings was significantly reduced by 25.03%, 33.48%, 46.26%, and 40.37%, respectively ($p < 0.05$), while the LDMC of the 'WY' and 'NX' seedlings significantly increased by 16.14% and 48.43%, respectively ($p < 0.05$). The SH of the 'YC' and the LL of the 'NX' seedlings were significantly reduced under stress conditions ($p < 0.05$), while the differences in growth characteristics among other provenances were insignificant.

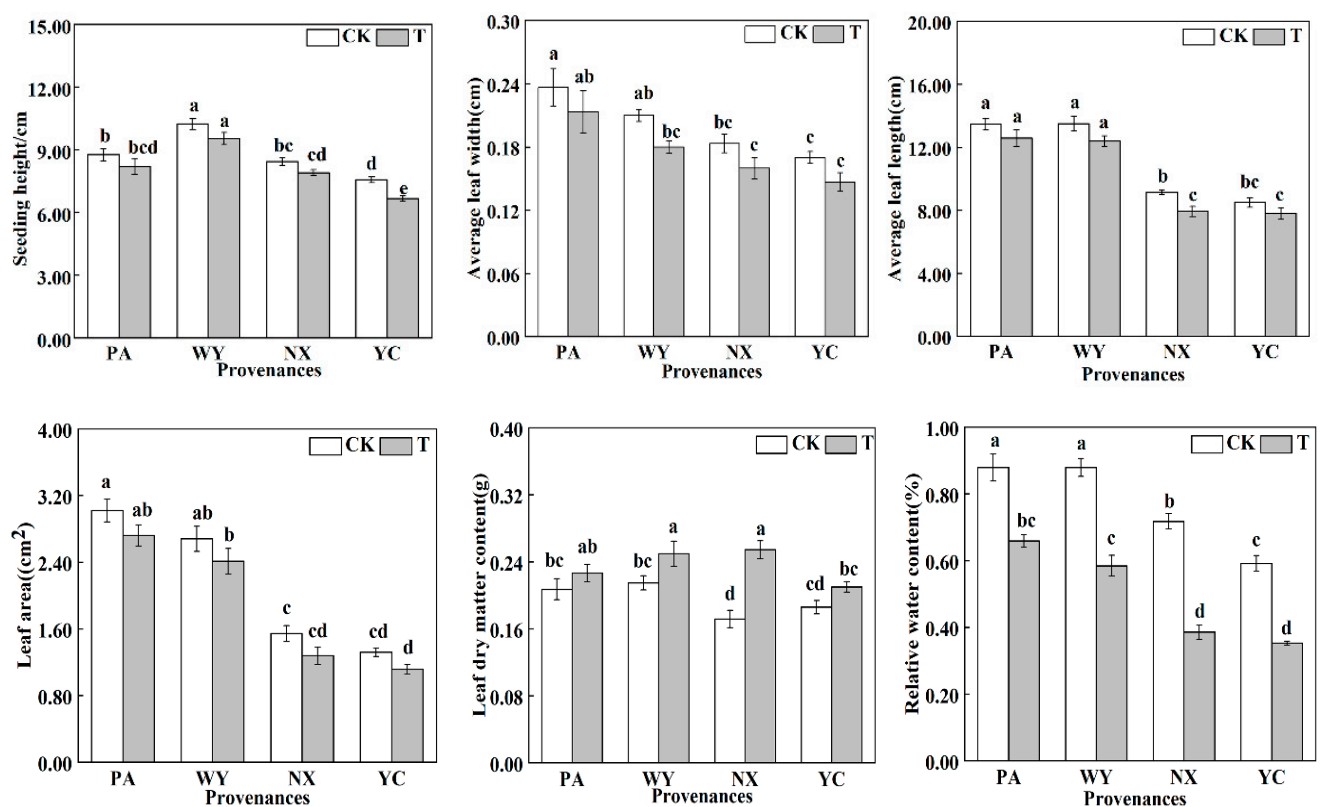

**Figure 3.** Effect of combined drought and low–temperature stress on seedling growth characteristics of different provenances. Values are means ± standard deviation (*n* = 3). Different small letters on the bars indicate significant differences between provenances ($p < 0.05$).

*3.4. Effects of Combined Drought and Low–Temperature Stress on Photosynthetic Characteristics of Different Seedlings*

3.4.1. Effects of Combined Drought and Low–Temperature Stress on Photosynthetic Gas Exchange Parameters of Different Seedlings

Under the non–stressed, the Pn of the 'YC' seedling was significantly higher ($p < 0.05$) than that of other seedlings ($p < 0.05$) (Figure 4). Under combined drought and low–temperature stress, the Pn of four seedlings was significantly reduced ($p < 0.05$), among which, the Pn of seedlings from the 'WY' and 'NX' decreased significantly by 86.47% and 80.06%, respectively. The Gs, Tr, and WUE of four seedlings were significantly reduced ($p < 0.05$) under combined drought and low–temperature stress. At the same time, Ci was significantly increased ($p < 0.05$), and the Gs and Tr of the 'NX' seedlings showed the most significant reduction compared with non–stressed, which were reduced by 74.72% and 73.41% ($p < 0.05$), respectively. Meanwhile, Ci of the 'NX' seedling suggested the most remarkable increase compared with non–stressed, which significantly increased by 83.67% ($p < 0.05$).

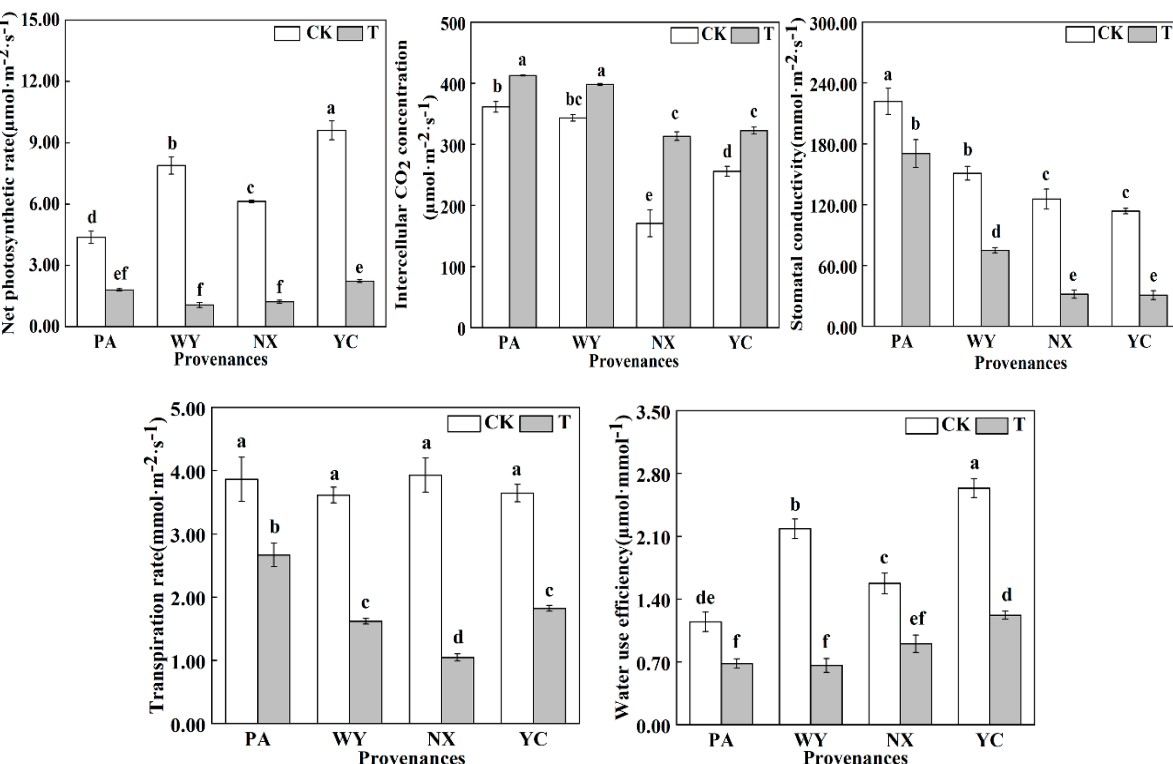

**Figure 4.** Effects of combined drought and low–temperature stress on photosynthetic gas exchange parameters of seedlings from different provenances. Values are means ± standard deviation (*n* = 3). Different small letters on the bars indicate significant differences between provenances ($p < 0.05$).

3.4.2. Effects of Combined Drought and Low–Temperature Stress on Chlorophyll Content of Different Seedlings

The Chla, Chlb, and Chla + b contents and Chla/b of different seedlings proved a decreasing trend under combined drought and low–temperature stress (Figure 5). Under the combined drought and low–temperature stress, the Chla/b of seedlings of the 'PA', 'WY', and 'NX' decreased compared with those of the control but did not differ significantly from those of the control. The Chla, Chlb, and Chla + b contents of seedlings from four provenances were reduced considerably compared to the control ($p < 0.05$). Among these, the Chlb and Chla + b contents of the 'WY' and NX seedlings decreased significantly by 40.74%, 43.17%, 39.14%, and 41.07%, respectively ($p < 0.05$).

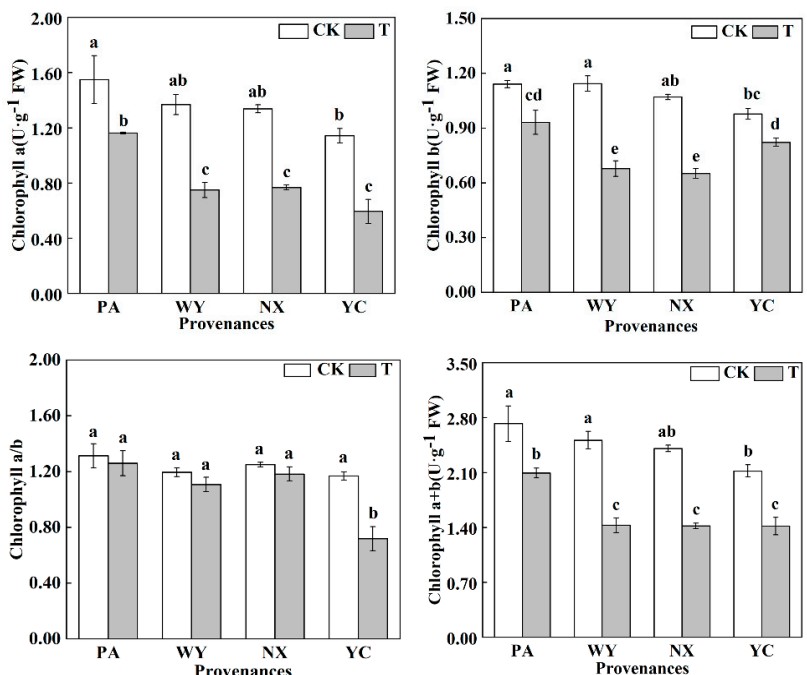

**Figure 5.** Effect of combined drought and low–temperature stress on chlorophyll content of seedlings from different provenances. Values are means ± standard deviation (*n* = 3). Different small letters on the bars indicate significant differences between provenances (*p* < 0.05).

*3.5. Effects of Combined Drought and Low–Temperature Stress on REC and MDA Content of Different Seedlings*

Under the non–stressed, there was no significant difference in REC among the different seedlings. In contrast, the REC of seedlings of the 'PA', 'WY', 'NX', and 'YC' were significantly increased under combined drought and low–temperature stress, which increased by 142.34%, 261.59%, 260.85%, and 192.23% compared to the non–stressed (*p* < 0.05) (Figure 6). The MDA content of seedlings from different provenances also significantly increased under stress (*p* < 0.05). Compared with the non–stressed, the 'PA' increased the least (32.50%), and the NX increased the most (111.64%). The results showed that the combined drought and low–temperature stress caused some damage to the cell membranes of the four provenances, and the damage to the 'PA' was the least.

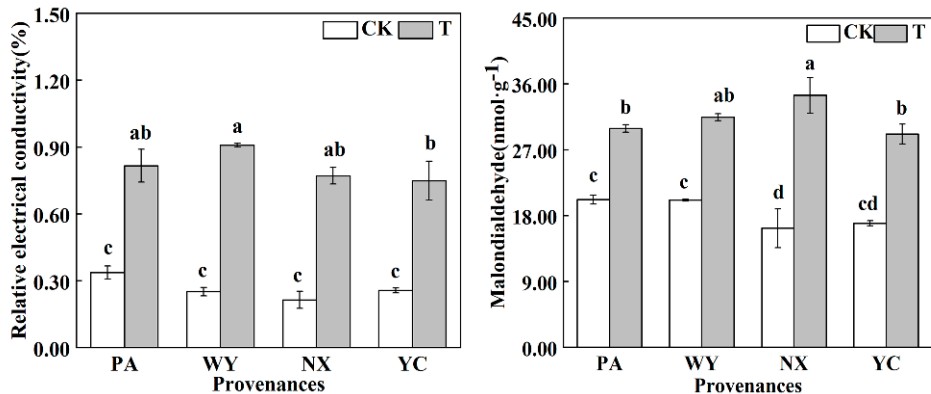

**Figure 6.** Effects of combined drought and low–temperature stress on REC and MDA content of seedlings from different provenances. Values are means ± standard deviation (*n* = 3). Different small letters on the bars indicate significant differences between provenances (*p* < 0.05).

### 3.6. Effects of Combined Drought and Low–Temperature Stress on Osmoregulatory Substances in Different Seedlings

Under the non–stress, the differences in the Pro content of seedlings from the 'PA', 'WY', and 'NX' were not significant, whereas the Pro content of seedlings from four provenances increased significantly under stress ($p < 0.05$) (Figure 7), with the tremendous increase (294.79%) in 'PA.' The SS content of four seedlings also increased significantly ($p < 0.05$) under the stress. The SS content of seedlings of the 'PA', 'WY', 'NX', and 'YC' increased by 2.17, 1.17, 1.36, and 1.50 times compared with the non–stressed, respectively.

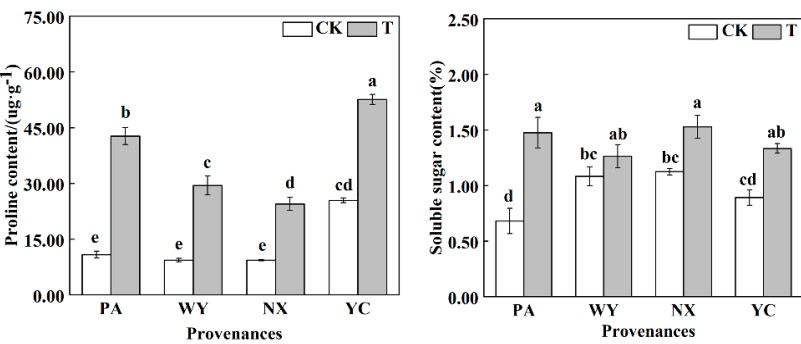

**Figure 7.** Effects of combined drought and low–temperature stress on osmoregulatory substances in seedlings of different provenances. Values are means ± standard deviation (*n* = 3). Different small letters on the bars indicate significant differences between provenances ($p < 0.05$).

### 3.7. Effects of Combined Drought and Low–Temperature Stress on Reactive Oxygen Levels in Different Seedlings

As shown in Figure 8, the differences in $O_2^{\bullet-}$ production rate of seedlings of the 'WY', 'NX', and 'YC' were not significant under the non–stressed, whereas the $O_2^{\bullet-}$ production rate of seedlings from four provenances were significantly increased under the combined drought and low–temperature stress ($p < 0.05$). Compared with the non–stressed, the $O_2^{\bullet-}$ production rate of seedlings of the 'PA', 'WY', 'NX', and 'YC' increased by 1.70, 2.05, 2.62, and 2.18 times, respectively. The $H_2O_2$ and $\cdot OH$ contents of four different seedlings also tended to increase under stress, among which the $H_2O_2$ and $\cdot OH$ contents of the 'PA' seedlings suggested the smallest increase. Compared with the non–stressed, the $H_2O_2$ and $\cdot OH$ contents of the 'NX' seedlings significantly increased by 26.44% and 95.30%, respectively ($p < 0.05$).

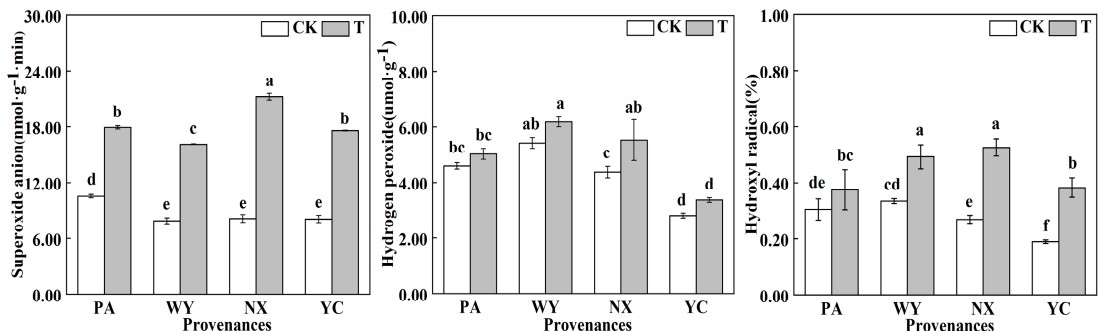

**Figure 8.** Effects of combined drought and low–temperature stress on reactive oxygen levels in seedlings of different provenances. Values are means ± standard deviation (*n* = 3). Different small letters on the bars indicate significant differences between provenances ($p < 0.05$).

### 3.8. Effects of Combined Drought and Low–Temperature Stress on Antioxidant Enzyme Activities of Different Seedlings

Under the non–stressed, the SOD activity of the 'YC' seedlings was significantly higher ($p < 0.05$) than that of other provenances ($p < 0.05$) (Figure 9). Under the combined drought

and low–temperature stress, the SOD activities of seedlings from all four provenances were significantly increased ($p < 0.05$), with a more significant increase in the SOD activities of seedlings from the 'PA' and 'YC', which increased by 256.31% and 204.16%, respectively. Under the non–stressed, both POD and APX activities of the 'NX' seedlings were significantly greater than those of other provenances ($p < 0.05$). Under combined drought and low–temperature stress, the POD, CAT, and APX activities of seedlings from four provenances were significantly increased ($p < 0.05$), with the greatest increase in seedlings from the 'PA', which increased by 353.99%, 465.71%, and 253.71% compared with the non–stressed, respectively. Furthermore, under stress, the increases in SOD, POD, CAT, and APX activities of the 'NX' seedlings were significantly lower than that of other provenances.

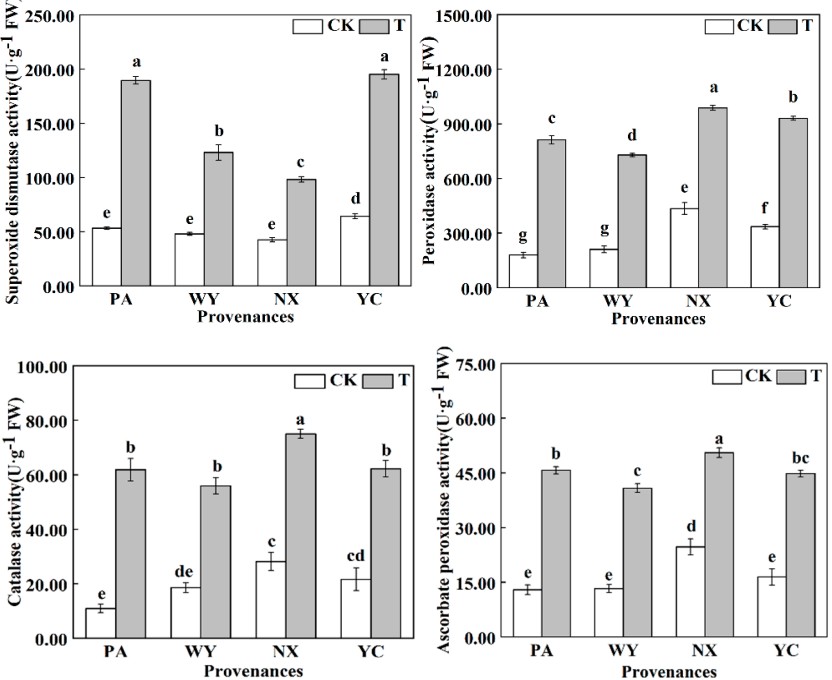

**Figure 9.** Effects of combined drought and low–temperature stress on antioxidant enzyme activities of seedlings from different provenances. Values are means ± standard deviation (*n* = 3). Different small letters on the bars indicate significant differences between provenances ($p < 0.05$).

### *3.9. Comprehensive Evaluation of Combined Drought and Low–Temperature Tolerance*
### 3.9.1. Principal Component Analysis (PCA)

Principal component analysis was used to evaluate the correlation between stress treatment and morphological, photosynthetic, and physiological and biochemical characteristics of *Poa annua* from different provenances. Four principal components (eigenvalues > 1) were extracted from PCA analysis, which explained 90.55% of the variability of the total data. With the first dimension (F1) having 50.67% of the variability and the second dimension (F2) having 25.13% (Figure 10A,B). The 26 traits divided the treatments and the *Poa annua* from four provenances into four quadrants (Q) of the PCA. Figure 10A proved a clear separation of the two treatments, where the stress group was in the right area of PC1 (Q3 and Q4), and the control group was on the left side of PC1 (Q1 and Q3). Similarly, we also found a significant separation of *Poa annua* from four provenances: the 'PA' and 'YC' were in Q1, while the 'NX' and 'WY' appeared in Q4 (Figure 10A). The biplot showed *Poa annua*'s morphological and photosynthetic parameters had a strong positive relationship. However, physiological and biochemical characteristics were exhibited in the opposite region, which indicated that the combined drought and low–temperature stress enhanced osmoregulatory capacity, ROS accumulation, and antioxidant enzyme activities and inhibited seedling growth.

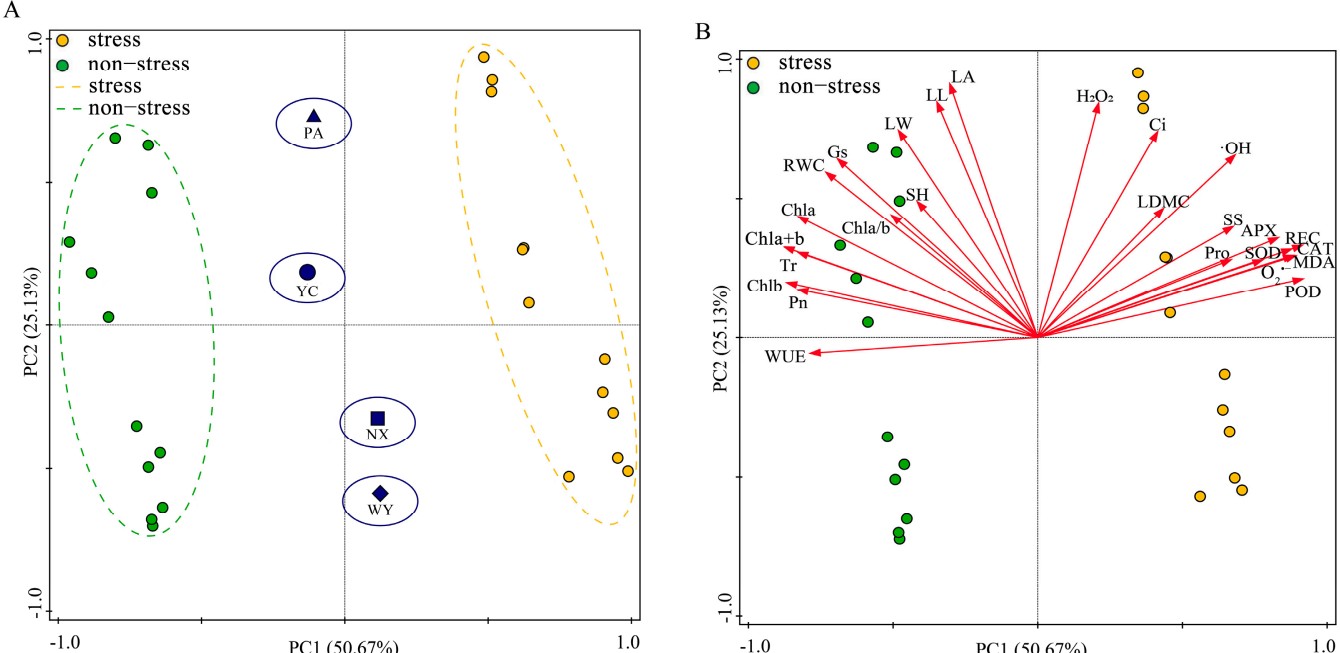

**Figure 10.** Observation plot showing separation of different treatments and *Poa annua* from different provenances in different quadrants (**A**). PCA biplot showing the correlations between morphological, photosynthetic, and physiological and biochemical characteristics of *Poa annua* seedlings under different combined water and temperature stress (**B**).

### 3.9.2. Membership Function Analysis

Based on the PCA screening results, membership function analysis assessed 20 traits of *Poa annua* seedlings from different provenances. The relative values of stress and non–stressed treatments were calculated, and the total average value was estimated to rank the combined drought and low–temperature resistance (Table 4). The most considerable value of the membership function was the 'PA' (0.977), followed by the 'YC' (0.386), whereas the membership function value of the 'WY' and 'NX' were 0.382 and 0.260, respectively. The results proved that the comprehensive order of combined drought and low–temperature tolerance of *Poa annua* seedlings from four provenances was 'PA' > 'YC' > 'WY' > 'NX', which corresponded to the morphological characteristics of seedlings from different provenances under combined drought and low–temperature stress and demonstrated the viability of the evaluation approach.

**Table 4.** Membership function values of *Poa annua* seedlings from four provenances under combined drought and low–temperature stress.

| Traits | Value of the Membership Function | | | |
|---|---|---|---|---|
|  | **PA** | **WY** | **NX** | **YC** |
| SH | 0.959 | 0.900 | 1.000 | 0.000 |
| LL | 1.000 | 0.763 | 0.000 | 0.735 |
| LW | 1.000 | 0.000 | 0.346 | 0.086 |
| LA | 0.665 | 1.000 | 0.000 | 0.158 |
| Chla | 1.000 | 0.140 | 0.235 | 0.000 |
| Chlb | 0.907 | 0.000 | 0.058 | 1.000 |
| Chla/b | 1.000 | 0.888 | 0.939 | 0.000 |
| Chla + b | 1.000 | 0.000 | 0.092 | 0.458 |
| Pn | 1.000 | 0.000 | 0.232 | 0.348 |
| Gs | 1.000 | 0.474 | 0.000 | 0.018 |
| Tr | 1.000 | 0.421 | 0.000 | 0.545 |

**Table 4.** *Cont.*

| Traits | Value of the Membership Function | | | |
| --- | --- | --- | --- | --- |
| | PA | WY | NX | YC |
| WUE | 1.000 | 0.000 | 0.911 | 0.546 |
| REC | 1.000 | 0.090 | 0.000 | 0.653 |
| MDA | 1.000 | 0.875 | 0.000 | 0.625 |
| Pro | 1.000 | 0.583 | 0.291 | 0.000 |
| SS | 1.000 | 0.000 | 1.102 | 1.157 |
| SOD | 1.000 | 0.189 | 0.000 | 0.580 |
| POD | 1.000 | 0.542 | 0.000 | 0.216 |
| CAT | 1.000 | 0.100 | 0.000 | 0.116 |
| APX | 1.000 | 0.675 | 0.000 | 0.488 |
| Average | 0.977 | 0.382 | 0.260 | 0.386 |
| Rank | 1 | 3 | 4 | 2 |

## 4. Discussion

Plants have adapted to the current growth conditions and evolved a wide range of morphological, physiological, and biochemical regulation mechanisms through long–term natural selection and genetic variation [41]. The 'PA' growing in high altitude, low–temperature, and rainfall areas had a stronger combined drought and low–temperature tolerance than the 'NX' growing in low altitude, high temperature, and rainy regions in the experiment, and the mechanism of this was mainly through the enhancement of antioxidant enzyme activities, the increase of osmoregulatory substances, and the reduction of membrane permeability and lipid peroxidation.

Drought and low–temperature stresses are the main abiotic factors affecting seedling growth and yield in Chinese agroecosystems [42,43]. The external morphological characteristics of plants can directly reflect the effects of combined drought and low–temperature stress on plants and thus can be used as an indicator of plant resilience [44,45]. By observing the external morphology of seedlings from four provenances after 24 h of combined drought and low–temperature stress, it was found that seedlings from the 'PA' had less severe damage symptoms than others, while the 'NX' had the most severe damage symptoms. Plant variation diversity is the result of the combined effects of genetic adaptation and environmental heterogeneity. The richer the variation of plant traits, the more they can improve their adaptability to different environments [46,47]. Our study found that under combined drought and low–temperature stress, the coefficient of variation of *Poa annua* varied from 8.75% to 78.40%, with an average coefficient of variation of 22.60%, which was much greater than 10% [48], indicating that the leaf traits were rich in variation, which was conducive to the screening of combined drought and low–temperature resistant materials. This may be the expression of some genes that are silenced at ambient and well–watered conditions induced by combined water and temperature stress and then come to cope with the damage of combined drought and low–temperature stress to the plant. Furthermore, this also provides the possibility for breeding specialists to find materials that meet the needs of production. The changes of plants LA, LL, LW, LW, SH, and LDMC also reflect the growth status and sensitivity of plants to adversity stress more intuitively and rapidly [49–51]. The growth indexes of LL, LW, LA, and SH of seedlings from four provenances were all inhibited under stress, in which the average growth indexes of the 'PA' decreased the least. Conversely, the variable 'NX' exhibits the highest frequency, which might be caused by provenance differences and various physiological changes. The LDMC of seedlings from four provenances increased under the combined water and temperature stress, indicating that four seedlings were tolerant; among them, the LDMC of the 'PA' seedlings increased the least, and that of the 'NX' seedlings increased the most, which indicated that *Poa annua* seedlings from different provenances had different growth strategies under stress. RWC is a key indicator reflecting the degree of hydration of plant cells and tissues, which is crucial for physiological and biochemical functions and growth processes [29]. Numerous

studies have shown that plants maintain higher RWC during stress, indicating stronger stress resistance [52–55]. The RWC of four seedlings was reduced under combined water and temperature stress, with the most negligible reduction in RWC from the 'PA' in this experiment (Figure 3). Thus, the 'PA' seedlings showed stronger combined drought and cold resistance in natural habitats. Moreover, our study also found that although the 'PA' had the lowest Pn under the non–stressed, the 'PA' maintained a higher Pn compared to the other provenances after 24 h of the stress, further indicating that the 'PA' exhibited stronger combined drought and low–temperature resistance.

When plants are subjected to abiotic stress, their photosynthetic gas exchange parameters are also affected. The photosynthetic rate can reflect the assimilation capacity of plants per unit leaf area and is an essential indicator for measuring plant photosynthetic capacity [56]. Our study proved that the Pn, Gs, Tr, and WUE of *Poa annua* seedlings from four provenances were significantly inhibited under stress, which was consistent with previous studies on rapeseed (*Brassica napus* L.) under combined drought and cold stress [57]. Interestingly, the decreases in Pn, Gs, Tr, and WUE of the 'PA' seedlings were much smaller than those of other provenances, suggesting that the 'PA' seedlings had stronger organic matter production capacity and higher leaf water potential under combined drought and low–temperature stress. Studies have shown that the causes of plant Pn changes are classified as stomatal factors and nonstomatal factors, which are mainly judged based on the consistency of the changes in Gs and Ci of the leaves; if the changes in Gs and Ci of plant leaves are consistent, then it is a stomatal factor, and if the changes in Gs and Ci are different, then it is a non–stomatal factor [58]. The Pn of the leaves of seedlings from four provenances decreased under stress in this experiment, accompanied by an increase in Ci, indicating that the limitation of non–stomatal factors was the main reason for the decrease in Pn of the leaves of different seedlings; this might be due to the temporary enhancement of cellular respiration under stress, which impaired the activity of photosynthetically related enzymes and the photosynthetic organs, leading to the decrease of photosynthesis in the leaves. Chlorophyll plays a key role in receiving and converting energy in a variety of plant metabolic activities under abiotic stress. The content of chlorophyll is considered to be an essential indicator for assessing plant growth and photosynthetic capacity. Meanwhile, Chlorophyll damage is closely associated with decreased plant productivity, growth, and stress tolerance [59–61]. The ultrastructure of plant chloroplasts is damaged, leading to the blockage of chlorophyll synthesis, as well as accelerated chlorophyll degradation under stress [62]. The results of this study showed that the Chla, Chlb, and Chla + b contents of the seedlings from four provenances were significantly inhibited after being subjected to combined drought and low–temperature stress, but the Chla, Chlb, and Chla + b contents of the 'PA' seedlings were still substantially higher than those of other provenances. This result confirmed that the 'PA' seemed to be better adapted to combined drought and low–temperature stress by protecting the chlorophyll from being decomposed. Relevant studies have shown that under stressful conditions, plants develop an imbalance in the photosynthetic light and dark reactions, and the photosynthetic electron transport chain releases more $O_2^{\bullet-}$ by transferring electrons to molecular oxygen, leading to poorer photosynthetic performance [63]. This poorer photosynthetic performance has been linked to changes in leaf pigment composition, reduced enzyme activity in the carbon cycle, and altered chloroplast development [64].

It is well known that Pro and SS play crucial roles in osmoprotection and thereby modulate plant tolerance when plants are subjected to stress [59,65,66]. The increase in Pro and SS contents is a common phenomenon observed in plants under stressful environments [67,68]. The Pro content of seedlings from provenances 'PA', 'WY', and 'NX' showed no significant difference under the non–stressed in our study, whereas the Pro and SS contents of seedlings from four provenances increased under combined drought and low–temperature stress, with the most significant increase in Pro and SS contents of the 'PA', indicating that the 'PA' had a stronger osmotic adjustment capacity. In this experiment, the combined drought and low–temperature stress caused a large accumulation

of MDA content, REC, $O_2^{\bullet-}$, $H_2O_2$, and ·OH contents in the leaves of *Poa annua*, and the accumulation in the 'NX' was larger than that in the other provenances, which suggests that the combined drought and low–temperature stress led to the membrane damage in the seedlings of four provenances and that the 'NX' seedlings had a poor combined drought and low–temperature resistance. There is an increased accumulation of ROS ($O_2^{\bullet-}$, $H_2O_2$, and ·OH) due to stress, leading to their attack on the cell membrane and oxidation, which results in an increase in lipid peroxidation and, in turn, increases the production of MDA and REC [69]. It is also noteworthy that under stress, ROS production and the balance between ROS production and scavenging lead to different cellular signaling events, such as cellular senescence, apoptosis, and cell signaling [70,71]. In order to limit the damage caused by ROS, plants can stimulate the supply of enzymatic and non–enzymatic antioxidants as a defense mechanism to cope with stressors [72–74]. Undoubtedly, antioxidant enzymes are another important member of the defense mechanism against oxidative damage caused by stress. SOD is an essential enzyme in the antioxidant system for converting $O_2^{\bullet-}$ to $O_2$ and $H_2O_2$. CAT can effectively convert toxic levels of $H_2O_2$ into $H_2O$ and $O_2$ [75]. Elimination of $H_2O_2$ by APX and CAT under combined stress is considered to be an effective scavenging system that improves abiotic stress tolerance in plants [76]. Our study found that the activities of SOD, POD, CAT, and APX increased to different degrees in the seedlings of four provenances under stress, which is considered a vital strategy to alleviate combined drought and low–temperature and induced oxidative stress. Among them, the 'NX' seedlings showed minor decreases in SOD, POD, CAT, and APX activities, while the 'PA' seedlings showed the most significant increases in POD, CAT, and APX activities, indicating that the 'PA' had stronger stress resistance, which may be related to the environment (soil, rainfall, temperature, etc.) of the provenance.

## 5. Conclusions

In the present study, we investigated the morphology, photosynthesis, and physiological and biochemical responses of *Poa annua* seedlings from four different provenances to combine drought and low–temperature stress. According to the results, combined drought and low–temperature stress were reflected in obvious impairment in photosynthesis, with inhibition of net photosynthetic rate, transpiration rate, stomatal conductance, and chlorophyll synthesis. ROS levels, MDA content, and REC were increased, highlighting oxidative damage to cell membranes and lipid peroxidation. However, the adverse effects of combined drought and low–temperature stress could be mitigated by various intracellular mechanisms, such as the accumulation of antioxidant enzymes (SOD, POD, CAT, and APX) and osmoregulatory substances (Pro and SS) (Figure 11). Combined drought and low–temperature stress induced different changes in *Poa annua* seedlings from four different provenances. Considering the results of principal component analysis and membership function, the comprehensive order of *Poa annua* from four different provenances according to their combined drought and low–temperature resistance was: 'PA' > 'YC' > 'WY' > 'NX', which was consistent with the phenotypic symptomatic performance of seedlings from the four provenances under stress, and demonstrated the variability of growth characteristics and physiological and biochemical traits under different combined stress.

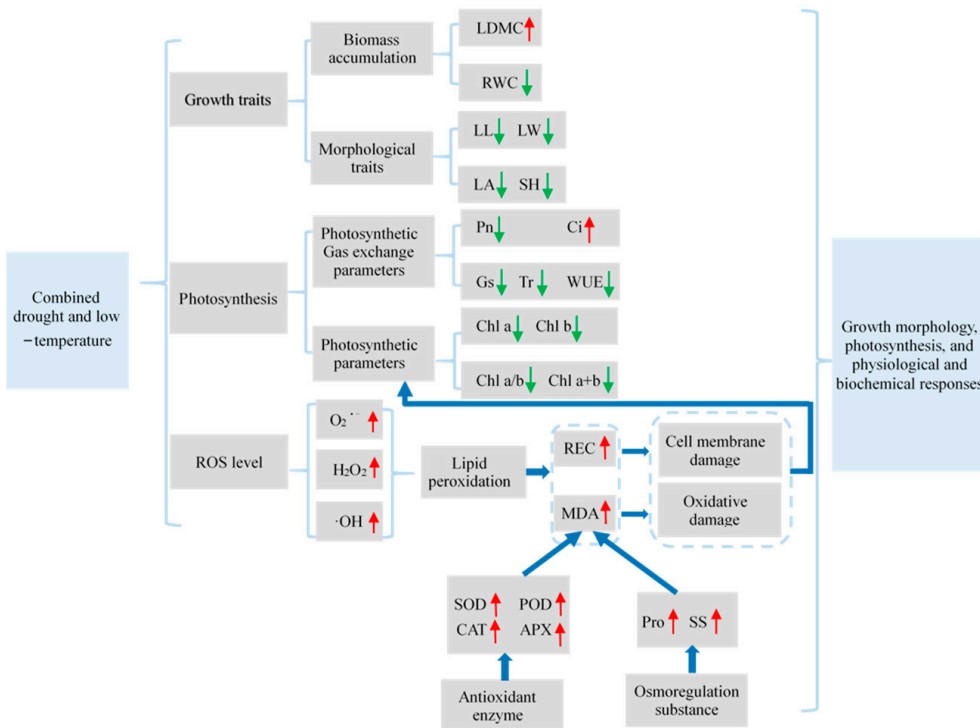

**Figure 11.** Schematic diagram of growth inhibition induced by combined drought and low–temperature stress. Red and green arrows indicate the increase and decrease of the corresponding traits under stress treatments, respectively.

**Author Contributions:** Conceptualization, J.L., X.B., F.R. and M.S.; formal analysis, X.B. and J.L.; investigation, J.L., F.R., P.L. and H.C.; supervision, X.B.; data curation, J.L.; writing—original draft, J.L.; writing—review and editing, F.R., M.S. and H.C.; project administration, X.B.; funding acquisition, X.B. All authors have read and agreed to the published version of the manuscript.

**Funding:** This work was supported by the National Natural Science Foundation of China (31560667); Gansu Forestry and Grassland Bureau Grassland Ecological Restoration and Management Science and Technology Support Project (GSLC-2020-3); "Innovation Star" Project for Outstanding Postgraduates in Gansu Province (2023CXZX-626).

**Institutional Review Board Statement:** Not applicable.

**Data Availability Statement:** The data are included in the article.

**Conflicts of Interest:** The authors declare no conflict of interest.

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
