# Peer review of "Photosynthetic and Physiological Responses to Combined Drought and Low–Temperature Stress in Poa annua Seedlings from Different Provenances"

_agriculture, doi:10.3390/agriculture13091781_

Round 1
Reviewer 1 Report
The authors have made a great effort in this research work, there are some comments joined to the highlights in the attachment file. Moreover, the general comments to the authors are: -
- This study needs to be supported with molecular identification of the plants from different provinces to build the discussion of different tolerance to the combination of drought and low temperature and the physiological aspects of the plants.
- The discussion in the present study lacks to suggest the necessary solutions and treatments to overcome the inability of plants to adapt to the factors of water stress and low temperature.

Author Response
List of Responses
Dear Editors and Reviewers:
Thank you for your letter and for the reviewers’ comments concerning our manuscript entitled ‘Photosynthetic and physiological responses to combined drought and low-temperature stress in Poa annua seedlings from different locations’ (Manuscript ID: agriculture-2574848). Those comments are all valuable and very helpful for revising and improving our paper, as well as the important guiding significance to our research. We have studied the comments carefully and have made a correction which we hope meets with approval. More detailed responses to the questions and concerns raised by reviewers are attached as follows. Each comment is quoted in italics and is followed by the corresponding detailed response in blue. All changes have been highlighted in yellow color in the revised manuscript.
Responds to reviewer’s comments
- The abstract is too long.
Authors’ Response:
Thank you for your constructive and helpful suggestions. According to your recommendation, we have streamlined the abstract section to make it more concise and clear.
- These conditions are related to any References or previous studies? (The cultivated substrate was farmland soil, sand, sheep manure, and organic nutrient soil (7:1:1:1; v/v). Each pot was filled with 1.8 kg of the mixed substrate, with a sowing rate of 8 g·m-2)
Authors’ Response:
Thank you for your constructive and helpful suggestions. The proportions of the cultivation substrates were based on previous researchers in the laboratory [1, 2], with slight modifications, and we have added relevant references in the revised version. Secondly, the weight of the mixed substrate contained in each pot was based on the actual size of the pot. Finally, the amount of sowing was determined based on previous field production experience and germination rate.
References:
[1] Yuan, Y. J.; Bai, X. M.; Zhu, Y. N.; Zhang, Y. J.; Yan, Y. B.; Zhang, C. Z.; Li, Y. J. Correlation between the rhizome expansion ability and endogenous hormones contents of wild Poa pratensis in Gansu Province. Chinese Journal of Eco-Agriculture 2021, 29(08):1359-1369. (In Chinese)
[2] He, J. Y. Studies on cold-resistance of nine varieties of wild Poa. M.D. Thesis, Gansu Agricultural University, Lanzhou, China, 2012. (In Chinese)
- How do you calculate the moisture content?
Authors’ Response:
Thank you for your constructive and helpful suggestions. We used the stratified soil sampling method to determine the field water holding capacity W1, and determined the soil water content W2 by weighing method. If we want to maintain the soil water content at 80% of the maximum water-holding capacity of the fields, the soil water content to be maintained W3 = W1 × 80%, and therefore, the amount of water to be replenished in the pots W = W3 × W4 (the dry soil weight of the pots). Furthermore, the method is described in the reference attached to line 132 in the revised version.
- Meaning is not clear, what do you mean by kill?
Authors’ Response:
Thank you for your constructive and helpful suggestions. After careful verification, we realized that the 'to kill' in the article itself is meaningless, and we have deleted it in line 142 of the revised version.
- What is the model, manufacturer, city, and country of oven used?
Authors’ Response:
Thank you for your constructive and helpful suggestions. According to your recommendation, we have made relevant supplements to the model, manufacturer, city, and country of the oven in the revised version.
- Name of instrument ( model, city and country of the manufacturer).
Authors’ Response:
Thank you for your constructive and helpful suggestions. According to your recommendation, we have made relevant supplements to the model, manufacturer, city, and country of the conductivity meter in the revised version.
- For all chemicals used in the manuscript , please identify the company, city, country and the grade.
Authors’ Response:
Thank you for your constructive and helpful suggestions. According to your recommendation, we have made relevant supplements to the company, city, country and grade of all chemicals in the revised version.
- Please identify the instrument used in heating and optical measurement. Is the centrifuge under cooling?
Authors’ Response:
Thank you for your constructive and helpful suggestions. According to your recommendation, we have added the names of the heating and optical measuring instruments to the revised version, as well as the model, manufacturer, city and country of the instrument. In addition, the heated mixture was cooled naturally and then centrifuged at 4 °C at 12,000 rpm/min for 10 min. We have made the corresponding modifications in the revised version.
- How do you calculate the level of MDA? If you are used the calibration stander curve. Also, the proline and soluble sugers.
Authors’ Response:
(1) Malondialdehyde (MDA) content was calculated as follows:
Standard curve: y=0.0391x-0.0076, R2=0.9997; x is the standard concentration (nmol/mL), y is ΔA.
MDA content (nmol/g)=(ΔA+0.0076)÷0.0391×VT÷W
=25.58×(ΔA+0.0076)÷W
Where, VT: Volume of extraction solution, W: Sample mass.
(2) Proline (Pro) content was calculated as follows:
Standard curve: y=0.0364x-0.0054,R2=0.9959 (x is the Proline content(μg/mL); y is ΔA).
Proline content(µg/g)=[(ΔA+0.0054)÷0.0364×V1]÷(W×V1÷VT)
=27.47×(ΔA-0.0047)÷W
Where, V1: Volume of sample in the reaction; VT: Volume of extract added; W: Sample mass.
(3) Soluble sugar (SS) content was calculated as follows:
Standard curve: y=0.0905x+0.5174,R2=0.995916 (x is Soluble sugar content (μg/mL);y is ΔA).
Soluble sugar content (%)=[(V1×ΔA×n)/(VT×W×106)]×100%
Where, V1: Volume of sample in the reaction;ΔA: Sugar content of extracts;n: Dilution ratio; W: Sample mass; VT: Volume of extract added.
- How do you calculate the superoxide radical? Identify the method of calculation and the formula used.
Authors’ Response:
(1) The formula and method for calculating superoxide radical (O2·-) are as follows:
Standard curve: y=0.0136x+0.0056, R2=0.9998; x is the standard concentration (nmol/mL), y is ΔA.
Superoxide anion content (nmol/g) = (ΔA -0.0056)÷0.0136×V1÷(V1÷VT×W) ×2
= 147.06×(ΔA-0.0056)÷W
Superoxide anion production rate (nmol/g·min) =147.06×(ΔA-0.0056)÷W÷T
= 7.353×(ΔA-0.0056)÷W
Where, VT: volume of extract added; V1: volume of sample in the reaction; W: mass of sample; T: Reaction time; 2: 2 molecules of O2·- participate in the reaction to produce 1 molecule of NO2-.
In addition, specific methods for the determination of O2·- have corresponding references in the revision.
- Do the same for highlights in lines 208 and 214. (Identify the method of calculation and the formula used)
Authors’ Response:
(1) The formula and method for calculating superoxide radical (H2O2) are as follows:
Standard curve: y=0.6674x-0.0155,R2=0.9997 (x is the standard concentration,μmol/mL;y is ΔA).
H2O2 content (μmol/g)= [(ΔA+0.0155)÷0.6674×V1]÷(W ×V1÷VT)
=2.25×(ΔA+0.0155)÷W
Where, VT: volume of extract added; V1: volume of sample in the reaction; W: mass of sample
(2) The formula and method for calculating hydroxyl radical scavenging (·OH) are as follows:
Hydroxyl radical scavenging (%) =1-(Adetermination-Ablank)÷(Acontrol-Ablank)×100%
In addition, specific methods for the determination of H2O2 and·OH have corresponding references in the revision.
- If this instrument had used in previous measurements, please but the identify in the first line which the instrument mentioned.(UV-visible spectrophotometer (Agilent Technologies Inc., Santa Clara, CA, USA).)
Authors’ Response:
Thank you for your constructive and helpful suggestions. According to your recommendation, we have identified the name of the instrument the first time it is used in the revised version (2.5 Determination of chlorophyll content), and have added the model, manufacturer, city and country of the instrument.
- 2.5 Statistical Analysis
Authors’ Response:
Thank you for your constructive and helpful suggestions. According to your recommendation, we have reorganized the section ‘2. Materials and Methods’ in the revised version to make it clearer.
- This study needs to be supported with molecular identification of the plants from different provinces to build the discussion of different tolerance to the combination of drought and low-temperature and the physiological aspects of the plants.
Authors’ Response:
Thank you for your constructive and helpful suggestions. The preliminary work of this study was to initially determine the tolerance of plants from different locations based on their phenotypic damage symptoms under combined drought and low-temperature stress. Then, based on the morphological and physiological responses of plants to combined drought and low-temperature stress, we further clarified the tolerance of plants from different locations, in order to screen out the plants with stronger combined drought and low-temperature resistance. Thank you very much for your valuable suggestion ‘needs to be supported with molecular identification of the plants from different provinces to build the discussion of different tolerance to the combination of drought and low-temperature and the physiological aspects of the plants’, which provides us with new ideas and references for our future research, and also points out the direction for our next research.
- The discussion in the present study lacks to suggest the necessary solutions and treatments to overcome the inability of plants to adapt to the factors of water stress and low-temperature.
Authors’ Response:
Thank you for your constructive and helpful suggestions. Currently, there are few studies on the effects of combined drought and low-temperature stress on the morphology and physiological and biochemical traits of Poa annua seedlings. the strength of plant stress resistance may also vary depending on the origin. In this context, we investigated the changes in morphological, physiological and biochemical indices of wild Poa annua seedlings from four locations under sufficient water (soil water content was 80% of the maximum water-holding capacity of the field) at room temperature (25 ℃) and combined drought (soil water content was 30% of the maximum water-holding capacity of the field) and low temperature (0 ℃). Our study aimed to compare the effects of combined drought and low-temperature stress on the growth of plants from different locations, and then to compare the combined drought and low-temperature tolerance of plants from different locations. We hoped to screen out locations with better growth effects so that they could be selected as materials for lawn establishment in drought-and cold-prone regions in the world. In addition, we have clarified the research aims of this study in the introduction of this paper. Therefore, in the discussion section, we focused on the combined drought and low-temperature resistance of different plants and evaluated and screened them.

Reviewer 2 Report
The authors undertook a comprehensive study, intertwining the impacts of two distinct abiotic stresses: drought and low-temperature. They meticulously examined the resulting effects across various factors encompassing morphological and growth characteristics, photosynthetic efficiency, chrophyll concentration, as well as contents of REC and MDA. Moreover, they delved into the realm of osmoregulatory substances, ROS levels, and the activities of antioxidant enzymes. To synthesize their findings, the researchers employed a diverse array of statistical methods, aiming to draw meaningful conclusions from this multifaceted dataset. While their efforts are commendable, certain aspects of their statistical analyses remain convoluted. For instance, the lack of citation for formulas (1) to (3) between L254 and L255 raises questions about their origin and appropriateness. Furthermore, within section 3.2, it is worth exploring whether any prior research has endeavored to amalgamate ANOVA analysis with CV, potentially shedding light on novel insights or methodologies.
Author Response
List of Responses
Dear Editors and Reviewers:
Thank you for your letter and for the reviewers’ comments concerning our manuscript entitled ‘Photosynthetic and physiological responses to combined drought and low-temperature stress in Poa annua seedlings from different locations’ (Manuscript ID: agriculture-2574848). Those comments are all valuable and very helpful for revising and improving our paper, as well as the important guiding significance to our research. We have studied the comments carefully and have made a correction which we hope meets with approval. More detailed responses to the questions and concerns raised by reviewers are attached as follows. Each comment is quoted in italics and is followed by the corresponding detailed response in blue. All changes have been highlighted in yellow color in the revised manuscript.
Responds to reviewer’s comments
- The lack of citation for formulas (1) to (3) between L254 and L255 raises questions about their origin and appropriateness.
Authors’ Response:
Thank you for your constructive and useful suggestions. We have added references to line 212 in the revised version.
- Within section 3.2, it is worth exploring whether any prior research has endeavored to amalgamate ANOVA analysis with CV. Potentially shedding light on novel insights or methodologies.
Authors’ Response:
Thank you for your constructive and helpful suggestions. There have been previous studies combining ANOVA analysis with CV (coefficient of variation), for example:
[1] Zhao, X. Q.; Zhao, C.; Niu, Y. N.; Chao, W.; He, W.; Wang, Y. F.; Mao, T. T.; Bai, X. D. Understanding and comprehensive evaluation of cold resistance in the seedlings of multiple maize genotypes. Plants 2022, 11.14:1881.
[2] Zhao, P. X.; Yang, X.; Yang, Z. L.; Tian, Z. X.; Yang, Y. X. Phenotypic variation and geographical differentiation of Lithocarpus litseifolius based on herbarium-specimen analysis, Acta Agric. Univ. Jiangxiensis 2023, 45(2):285-297.(In Chinese)
- Potentially shedding light on novel insights or methodologies (Within section 3.2, it is worth exploring whether any prior research has endeavored to amalgamate ANOVA analysis with CV).
Authors’ Response:
Thank you for your constructive and beneficial suggestions. The combination of ANOVA analysis and CV may reveal new insights, and we have made relevant additions in the "Discussion" section of the revised version.
Analysis of variance (ANOVA) showed that the resistance of Poa annua to combined drought and low-temperature stress was controlled by its genetic composition, combined water and temperature stress, and the interaction of major factors. Genetic differences were confirmed by the genetic coefficient of variation of Poa annua among different locations when it was controlled by combined temperature and moisture at the seedling stage. In this study, the coefficient of variation of a Poa annua under combined drought and low-temperature stress ranged from 8.75% to 78.40%, with an average coefficient of variation of 22.60%, which was much greater than 10%, indicating that the leaf traits were rich in variation, which was favorable for the screening of combined drought and low-temperature resistant materials. This may be the expression of some genes that are silenced at ambient and well-watered conditions induced by combined temperature and water stress, and thus come to cope with the damage of combined drought and low-temperature stress to the plants. In addition, this also provides the possibility for breeding specialists to find materials that meet the needs of production.

Reviewer 3 Report
My contributions and suggestions are described in the pdf manuscript

Author Response
List of Responses
Dear Editors and Reviewers:
Thank you for your letter and for the reviewers’ comments concerning our manuscript entitled ‘Photosynthetic and physiological responses to combined drought and low-temperature stress in Poa annua seedlings from different locations’ (Manuscript ID: agriculture-2574848). Those comments are all valuable and very helpful for revising and improving our paper, as well as the important guiding significance to our research. We have studied the comments carefully and have made a correction which we hope meets with approval. More detailed responses to the questions and concerns raised by reviewers are attached as follows. Each comment is quoted in italics and is followed by the corresponding detailed response in blue. All changes have been highlighted in yellow color in the revised manuscript.
Responds to reviewer’s comments
- Specify better (different provenances); Suggestion: Different locations.
Authors’ Response:
Thank you for your constructive and helpful suggestions. According to your recommendation, We have modified the title in the revised version.
- I think that here it would be important to highlight the influences of cold on metabolism and of drought, then put the common points. I missed talking about enzymatic synetics since low temperatures will interfere with the carboxylation of RuBisCO.
Authors’ Response:
Thank you for your constructive and helpful suggestions. This experiment did not involve single drought and low-temperature stresses in the interim, therefore, the effect of a single stress on the plants was not described. However, thank you very much for your valuable recommendation, which pointed out the direction for our later research. Meanwhile, according to your recommendation, regarding the enzymatic response of plants to stress, we have made further improvements in the revised version to make it more explicit.
- locality
Authors’ Response:
Thank you for your constructive and helpful suggestions. According to your recommendation, we have modified relevant content about ‘locality’ in the revised version.
- Describe the treatments in detail (2.2. Stress treatment).
Authors’ Response:
Thank you for your constructive and helpful suggestions. According to your recommendation, we have added details of the experimental treatments in the revised version.
- When they presented 6-7 ...
Authors’ Response:
Thank you for your constructive and helpful suggestions. Poa annua is a two-month-old seedling that presents 6-7 true leaves. In order to make the timing of plant treatments readily apparent, we have modified the description in the revised version.
- Very low number of repetitions! these variables tend to show a lot of variation so it is important to have at least 5 repetitions to be sure of the performance.
Authors’ Response:
Thank you for your constructive and helpful suggestions. Undoubtedly, three replicates were indeed relatively few in the experiment, but to some extent, they can also indicate the problem. Thank you very much for your valuable recommendation. We will increase the number of replications in subsequent experiments to improve the reliability of the results.
- I think that to achieve your research objective you should have treatments where isolated stresses are applied so that you can understand the mechanisms involved in this response. Therefore, I suggest that you be careful in the next experiment, adding these treatments.
Authors’ Response:
Thank you for your constructive and helpful suggestions. Our main work in the early stage was to screen out the two materials with the strongest and most sensitive combined drought and low-temperature resistance, and then explore the responses of these two materials under single drought, low-temperature and combined drought and low-temperature stress, respectively, to further understand the corresponding mechanisms involved. Thank you very much for your valuable recommendation, which provides us with reference and firm motivation for our subsequent research and point out the direction for our next research.
- Figure titles must be self-explanatory. insert the description of the acronyms of the treatments.
Authors’ Response:
Thank you for your constructive and helpful suggestions. According to your recommendation, we have revised the relevant content in the revised version.
- Describe in table title(Factorial ANOVA of the provenance and combined drought and low-temperature stress, 303 their interactions, and genetic variation coefficient on seedlings from four provenances.)
Authors’ Response:
Thank you for your constructive and helpful suggestions. According to your recommendation, we have revised the relevant content in the revised version.
- Unnecessary (3.4.1. Effects of combined drought and low-temperature stress on photosynthetic gas exchange parameters of different seedlings)
Authors’ Response:
Thank you for your constructive and helpful suggestions. The measurement of photosynthetic gas exchange parameters is an important method to study the photosynthetic performance of plants, to diagnose the operation of their photosynthetic mechanisms, and to study the effects of environmental factors on photosynthesis. It has also been shown in related studies [1, 2] that the photosynthetic capacity of plants directly affects plant growth, biomass and stress tolerance, which is mainly reflected in the indexes of net photosynthetic rate (Pn), intercellular CO2 concentration (Ci) and stomatal conductance (Gs). Therefore, photosynthetic gas exchange parameters can also indicate the strength of plant stress tolerance to a certain extent.
[1] Kate M, Johnson G N. Chlorophy II fluorescence- a practical guide[J] J. Exp. Bot. 2000, 51(345): 659-668.
[2] Dou, X. Y.; Wu, G. J.; Huang, H. Y.; Hou, Y, J.; Gu, Q.; Peng, C. L. Responses of Jatropha curcas L. seedlings to drought stress. Chin. J. Appl. Ecol. 2008, 19(7):1425-1430.
- Please discuss the results in a more integrated way, in order to explore the physiological processes involved.
Authors’ Response:
Thank you for your constructive and helpful suggestions. According to your recommendation, we have further discussed the results comprehensively in the revised version.

Round 2
Reviewer 1 Report
The authors adequately answered all my comments and I recommend accepting the manuscript for publication in Agriculture.
Reviewer 3 Report
The authors made the necessary adjustments, so the article "Photosynthetic and physiological responses to combined drought and low-temperature stress in Poa annua seedlings from different locations" has merit for publication.